# Diffusion Autoencoders with Perceivers for Long, Irregular and Multimodal Sequences

## Abstract

Self-supervised learning has become a central strategy for representation learning, but the majority of successful architectures assume regularly-sampled inputs such as images, audios. and videos. In many scientific domains—e.g., astrophysics, data arrive instead as long, irregular, and multimodal sequences which existing methods struggle to natively support. We introduce the Diffusion Autoencoder with Perceivers (daep), a diffusion autoencoder architecture designed for such settings. Our method tokenizes heterogeneous measurements, compresses them with a Perceiver encoder, and reconstructs them with a Perceiver-IO diffusion decoder, enabling scalable learning in diverse data settings. We also adapt masked autoencoders (MAE) with a Perceiver architecture, establishing a strong baseline in the same architectural family. Across spectroscopic, photometric, and multimodal astronomical datasets, daep achieves lower reconstruction error and produces smoother, more discriminative latent spaces than VAE and perceiver-MAE baselines, particularly when preserving high-frequency structure is critical for downstream objectives. These results establish daep as an effective framework for scientific domains where data arrives as irregular, heterogeneous sequences.

## 1 Introduction

Self-supervised learning (SSL) has emerged as a powerful paradigm for representation learning, driving major advances in language, vision, and audio domains (Jing & Tian, 2020). These successes, however, typically rely on data defined on regular grids—for example, pixels in images or fixed-rate samples obtained in audio and video.

In many scientific and real-world applications, data instead arrive as long, irregularly sampled sequences. Biomedical records contain patient measurements collected at uneven intervals (Krishnan et al., 2022), financial markets respond to discrete and unpredictable news events, and in astrophysics, and both photometric and spectroscopic observations in astrophysics are obtained on irregular grids due to observational constraints. Developing SSL methods that can natively handle such irregular, multimodal inputs is therefore an important open challenge across domains.

Astronomical surveys provide an especially demanding benchmark for this setting. Large-scale projects such as SDSS (York et al., 2000), DESI (Abareshi et al., 2022), ATLAS (Tonry et al., 2018), ZTF (Bellm et al., 2018; Masci et al., 2018; Graham et al., 2019; Dekany et al., 2020), and YSE (Jones et al., 2021; Aleo et al., 2023) deliver petabyte-scale, irregularly sampled measurements of time-varying phenomena across multiple modalities (images, light curves, and spectra). These datasets are widely used in astrophysics (Zhang et al., 2024; Rizhko & Bloom, 2025) and provide a natural stress-test for scalable SSL architectures on irregular, multimodal data.

Reconstruction-based SSL has proven particularly effective in recent years. Diffusion models achieve state-of-the-art sampling in the image domain (Ho et al., 2020; Dhariwal & Nichol, 2021), but their latent spaces can be elusive (Kwon et al., 2022). Diffusion autoencoders (Preechakul et al., 2022) address this by coupling an encoder with a diffusion decoder, producing both meaningful features and high-quality reconstructions. However, existing approaches are tailored to regularly-sampled modalities such as images and do not transfer directly to irregular, long, multimodal sequences.

In this paper we introduce the Diffusion AutoEncoder with Perceiver (daep), a architecture designed for self-supervised learning of long, irregular, and multimodal sequences. Our proposed architecture combines three components: (i) a Perceiver encoder that flexibly handles variable-length tokenized inputs across modalities, (ii) a compact latent bottleneck for representation learning, and (iii) a Perceiver-IO diffusion decoder that reconstructs the original sequence without assuming a regular grid. This design allows daep to scale to datasets with millions of heterogeneous data samples, while producing both high-fidelity reconstructions and semantically-structured latent spaces.

To contextualize our approach, we also develop a masked autoencoder (MAE) baseline using the same Perceiver backbone, enabling a controlled comparison between masking-based and diffusion-based objectives in the irregular-sequence regime. Across spectroscopic, photometric, and multimodal astronomical datasets, daep achieves comparable or lower reconstruction error and stronger downstream classification performance than VAE and Perceiver-MAE baselines, with particular improvements in reconstructing critical high-frequency data features. While motivated by astrophysical data, the architecture is domain-agnostic and can be used for representation learning in healthcare, finance, and other areas where irregular multimodal sequences are frequently obtained.

## 2  BACKGROUND

**Diffusion models.** Diffusion models are score-based generative models that achieve state-of-the-art performance in domains such as image and video generation. These models learn a gradual denoising process that transforms pure noise into data. Generation proceeds by removing Gaussian noise drawn from the prior $\mathcal{N}(0, I)$ into a clean data sample after $T$ denoising steps. Ho et al. (2020) proposed to learn a noise model $\epsilon_\theta(x_t, t)$ that predicts the noise added at diffusion time $t$ to the corrupted data $x_t$. The model is trained by minimizing $||\epsilon_\theta(x_t, t) - \epsilon_t||$, where $\epsilon_t$ is the true noise added to clean data $x_0$ to produce $x_t$. During generation, the corruption process is inverted to produce a trajectory $x_T, x_{T-1}, \ldots, x_0$, typically with large $T$. Song et al. (2020) introduced a deterministic variant, the denoising diffusion implicit model (DDIM), which enables generation in fewer steps using the trained noise prediction model. When conditioning variables $z$ are available, the noise model can be extended as $\epsilon_\theta(x_t, z, t)$ and trained with the same loss $||\epsilon_\theta(x_t, z, t) - \epsilon_t||_2^2$.

**Diffusion autoencoders.** Diffusion autoencoders were originally proposed for the image domain by Preechakul et al. (2022). They encode data and reconstruct it using a conditional diffusion model. Because the encoding guides every denoising step, diffusion autoencoders capture fine-grained detail more effectively than, for example, variational autoencoders (Kingma & Welling, 2013). However, the original design relied on U-Nets (Preechakul et al., 2022; Dhariwal & Nichol, 2021), which are better suited to regular modalities such as images. Formally, the model encodes tokenized data into a latent representation $z = \text{Enc}_\theta(x)$, and a conditional score model $\epsilon\theta(x_t, z, t)$ decodes data via a diffusion process. Training minimizes the score-matching loss $||\epsilon_\theta(x_t, \text{Enc}_\theta(x), t) - \epsilon_t||_2^2$.

**Perceiver.** Perceiver and Perceiver-IO (Jaegle et al., 2021b;a) provide a general framework to (1) encode irregularly sampled sequences into a latent representation and (2) query outputs from this latent space. This makes them a natural fit for integration with diffusion transformers (Dhariwal & Nichol, 2021), enabling scalable representation learning with diffusion autoencoders.

**Masked autoencoders.** Masked autoencoders (MAEs) (He et al., 2022) are another approach to representation learning with an autoencoding loss. Instead of reconstructing the full data from the latent alone, MAEs mask a subset of the input and decode conditioned on both the latent representation and the unmasked portion. Formally, the data are split into two parts: a masked portion $x_\text{m}$ and an unmasked portion $x_\text{u}$. The latent representation is obtained by encoding the unmasked portion, $z = \text{Enc}_\theta(x_\text{u})$, and the decoder learns a function $\text{Dec}_\theta(x_\text{u}, z)$ by minimizing the masked reconstruction loss $||x_\text{m} - \text{Dec}_\theta(x_\text{u}, \text{Enc}_\theta(x_\text{u}))||_2^2$. This model is not a full decoder as in daep since it needs access to the portions of the measurements we would like to reconstruct rather than having only access to e.g., positional information.

**Related work.** Autoencoding and dimensionality reduction have a long history in representation learning. Early models that remain widely used include variational autoencoders (VAEs; Kingma & Welling, 2013) and their variants, such as hierarchical VAEs (Vahdat & Kautz, 2020) and Vector-Quantized VAEs (Van Den Oord et al., 2017; Razavi et al., 2019). These models are still common in physics applications, though they often suffer from posterior collapse (Van Den Oord et al., 2017;

Higgins et al., 2017) and are generally less expressive than GANs (Goodfellow et al., 2020) or diffusion models (e.g., DDPM; Ho et al., 2020).

Researchers have explored combining VAEs with diffusion models to improve generative quality, for example by learning a diffusion prior (Wehenkel & Louppe, 2021) or training diffusion models on VAE latent spaces (Kwon et al., 2022; Yan et al., 2021). Beyond VAEs, masked autoencoders (MAEs; He et al., 2022) have recently gained attention as efficient learners for images and videos, but primarily for regularly sampled modalities. MAEs reconstruct masked regions from unmasked context, a strategy well suited to modalities with strong local structure such as images or audio (and diffusion models can themselves be interpreted as a form of MAE; Wei et al. 2023); this strategy is less effective for data with long-range dependencies, particularly where data is sparse. Despite their impressive performance in image and audio domains, these methods also struggle to encode high-frequency structure in irregularly-sampled sequences with large number of tokens.

## 3 DIFFUSION AND MASKED AUTOENCODER WITH PERCEIVER

In this section, we introduce our **d**iffusion **a**uto**e**ncoder with **p**erceiver (daep[1]) and the corresponding masked autoencoder. A unimodal daep has three components: a tokenizer, an encoder, and a diffusion decoder. The corresponding masked autoencoder has a tokenizer, an encoder, and a direct decoder.

**Tokenizers.** We represent raw data as a sequence of tokens in the model dimension. Data are treated as a collection of measurements at specific locations with accompanying metadata. Formally, we define $(v, s, m)$, where $v$ denotes measurement values (e.g., the flux of an astrophysical source), $s$ provides positional information (e.g., wavelength, time, or photometric filter), and $m$ encodes observational metadata (e.g., which telescope was used, observation time of spectra). We adapt the perceiver strategy (Jaegle et al., 2021b) by linearly projecting $v$, using fixed sinusoidal embeddings followed by a small MLP for continuous parts of $s$ (e.g., time), inspired by Peebles & Xie (2023); and categorical embeddings for discrete parts (e.g., photometric filters). We concatenate value and positional embeddings and project them to the model dimension. Metadata are represented as extra tokens and appended to the sequence.

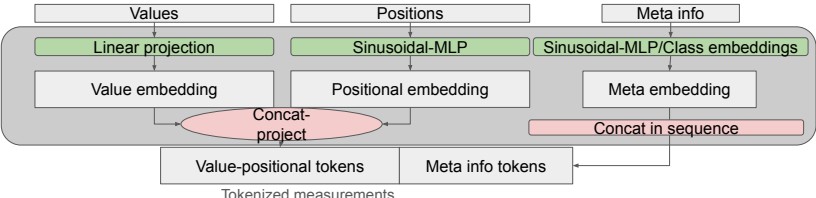

Figure 1: Schematic of tokenizers for general irregularly measured sequences.

**Unimodal encoders for both diffusion and masked decoders.** We use perceiver encoders (Jaegle et al., 2021b) to map token sequences into compact bottleneck representations, denoted $\text{Enc}_\theta$. Input tokens act as Keys and Values in cross-attention, while bottleneck representations serve as Queries. Self-attention is applied only among bottleneck sequences. We repeat these perceiver blocks several times, optionally sharing weights. This design handles variable-length sequences with linear cost in sequence length, making it efficient for processing long and irregular data. Finally, we project bottleneck sequences from the model dimension to a fixed bottleneck dimension. We illustrate the encoder in fig. 2. Since perceivers do not require fixed-length input, they can process both masked inputs for MAE training and full data for daep training.

**Perceiver-IO–based decoder.** Our diffusion decoder builds on diffusion transformers (Peebles & Xie, 2023), particularly cross-attention conditioning. Diffusion time is encoded with fixed sinusoidal embeddings passed through an MLP, as in Peebles & Xie (2023), and concatenated with the conditioning representation for the score model. The score model $\epsilon_\theta(x_t, z, t)$, which predicts added noise, is a perceiver-IO: noisy data are tokenized, concatenated with conditioning tokens, and used as Keys and Values in cross-attention. A latent sequence serves as Queries with self-attention, then

---

[1]Code available here.

acts as Keys and Values in a second cross-attention stage with positional information as Queries. We repeat these blocks, optionally sharing weights. The schematic is shown in fig. 2. While Jaegle et al. (2021a) recommend latent lengths of 128–512, this may exceed the input sequence length in some tasks. In such cases, we use a single-stage perceiver decoder without a latent sequence, directly connecting noisy tokens to noise prediction through cross-attention.

Because the perceiver-IO architecture is agnostic to both input and query length, the same decoder can be used for the masked autoencoder task as $\text{Dec}_\theta(x_{\text{umsk}}, z)$. In this case, we input the tokenized full data sequence with masked locations replaced by a learnable mask token, concatenate with the conditioning $z$ (but no diffusion time $t$), and query using positional information from masked locations only.

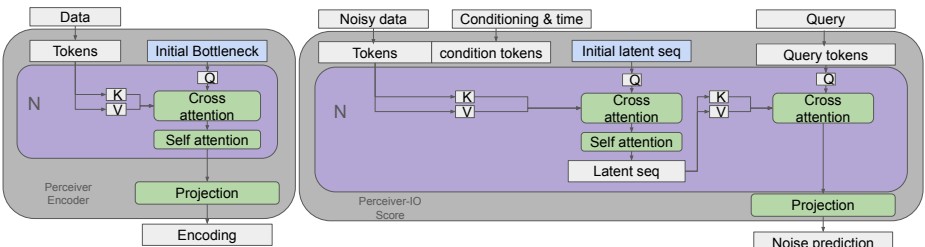

Figure 2: Schematic of the perceiver encoder and perceiver-IO score/decoder model used in daep and mae.

**Training and sampling for daep.** We train with the score-matching loss $||\epsilon_\theta(x_t, \text{Enc}_\theta(x_0), t) - \epsilon_t||_2^2$ from DDPM (Ho et al., 2020), using 1,000 denoising steps. At inference time, we adopt deterministic DDIM (Song et al., 2020) for faster sampling with 200 steps. Similar to Preechakul et al. (2022), our model is not inherently generative, since it requires the bottleneck representation of the input data for conditioned decoding. However, following Preechakul et al. (2022) and Wehenkel & Louppe (2021), we can train another DDIM to sample from the bottleneck distribution, enabling prior generation.

**Training the masked autoencoder.** We train with a masked reconstruction loss $||x_{\text{msk}} - \text{Dec}_\theta(x_{\text{umsk}}, \text{Enc}_\theta(x_{\text{umsk}}))||_2^2$. In experiments, we train two variants with different masking ratios of the input: mae-75% masks 75% of measurement values, and mae-30% masks 30% of measurement values during training. For reconstruction tasks, we provide 10% of the tokens as unmasked to ensure a relatively fairer comparison with daep and VAEs that does not have access to raw measurements.

**VAE baselines.** For benchmarking, we train a $\beta$-VAE ($\beta = 0.1$) with the same perceiver encoder and decoder as both daep and mae. The benchmark models are trained with a weighted sum of the KL-divergence and the L2 reconstruction loss.

## 4 UNIMODAL EXPERIMENTS

### 4.1 HIGH-RESOLUTION SPECTRA OF VARIABLE STARS.

**Data source.** We used data from v2.0 DR9 of the Large Sky Area Multi-Object Fiber Spectroscopic Telescope (LAMOST; Cui et al., 2012), specifically the dataset consolidated by Rizhko & Bloom (2025). The dataset contains spectra of variable stars with an average of ~2,500 flux measurements, and a maximum of ~4,000 per star. In total, we use 17,063 spectra for training and 2,225 for testing. Full architectural details are provided in appendix A.2.

**Reconstruction.** In this task, we let the model reconstruct the observed data. For MAE models, we encode the full observed data and provide 10% of unmasked tokens during decoding. We show two enlarged test examples in fig. 3, with additional examples in fig. 12. Quantifying residual distributions as a function of wavelength is non-trivial because spectra are not aligned on a uniform grid. Instead, we plot all test residuals in fig. 4, with summary metrics in table 1. Both daep and MAE with perceivers achieve superior reconstruction than the baseline $\beta$-VAE, with daep showing fewer residuals in lower-wavelength regions and capturing finer spectral features. Interestingly, for

these long sequences, VAEs—even with small $\beta$ values—tend to reproduce only the low-frequency stellar continuum, while daep and mae recover higher-frequency structure in the data.

**Downstream classification task.** We classify variable stars into ten classes using linear probing on 30% of the test set. For MAE, we unmask all measurements during probing. Accuracy and $F_1$ scores are reported in table 1. Both daep and MAE outperform VAE, with daep achieving the highest $F_1$ score, indicating stronger representation learning.

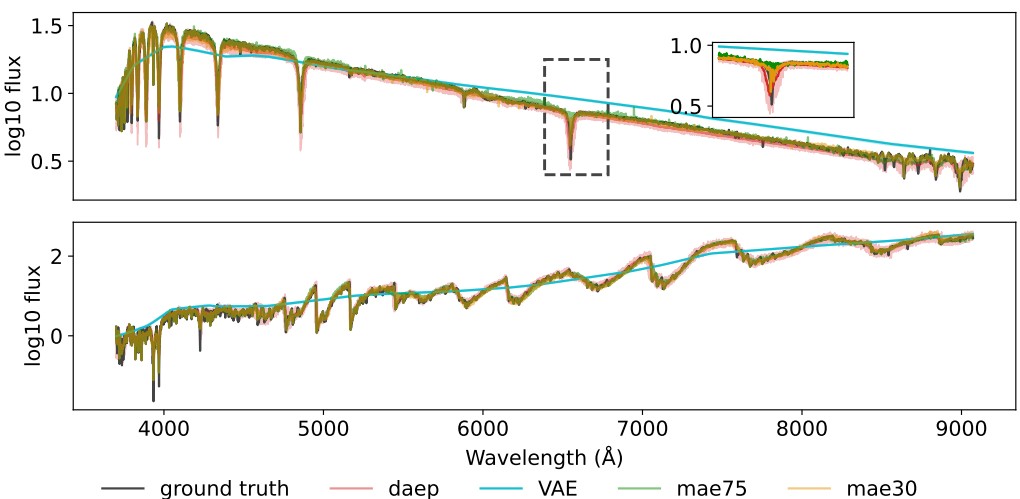

Figure 3: Two example reconstructions of variable star spectra. daep captures finer spectroscopic features, while the VAE mainly reproduces the continuum, likely due to posterior collapse. Both daep and MAE successfully capture small-scale spectral features.

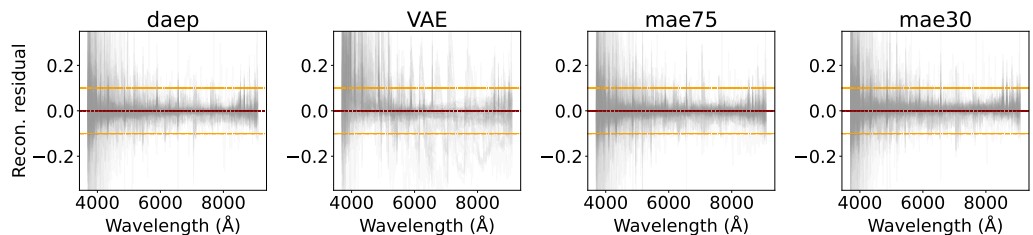

Figure 4: Residuals for test spectra in the LAMOST dataset. Both daep and MAE show smaller residuals than VAE. daep also preserves higher-resolution structure, with fewer residual spikes (e.g., near 7000 Å) compared to MAE. The red line marks 0 residual while the orange line marks a residual of ±0.1.

| Method | Abs. reconstruction error ↓ | Linear probing Accu. ↑ | Linear probing $F_1$ ↑ |
|---|---|---|---|
| daep | 0.038 (0.034) | **0.57 (0.003)** | **0.25 (0.003)** |
| VAE | 0.076 (0.075) | 0.54 (0.002) | 0.24 (0.004) |
| mae-75% | 0.036 (0.034) | 0.55 (0.002) | 0.23 (0.002) |
| mae-30% | **0.036 (0.029)** | 0.56 (0.002) | 0.23 (0.003) |

Table 1: Reconstruction and downstream linear probing metrics on LAMOST spectra. Best-performing models are boldfaced; underlined results indicate models whose 1,std interval overlaps with the best mean.

## 4.2 LOW-RESOLUTION SPECTRA OF SUPERNOVAE.

**Data source.** Next, we use data from the Zwicky Transient Facility Bright Transient Survey (ZTF-BTS; Bellm et al., 2018). The spectra were obtained from multiple different facilities and corrected for redshift, resulting in irregular grids in measurement position (wavelength). We encoded the spectra into four latent tokens of dimension four.

**Reconstruction.** As before, we let the model reconstruct the observed data. For MAE-based models, we encode the full observed data and provide 10% of unmasked tokens during decoding. We provide two test examples in fig. 5, with additional examples in fig. 14. The population-level reconstruction error is visualized in fig. 6. For this task, daep produces smaller residuals than both VAE and MAE models across the full wavelength range, consistent with the metrics in table 2. This may be due to the diffusion decoder's ability to better handle noise and data artifacts, which contaminate the low-resolution ZTF spectra more severely than the LAMOST spectra.

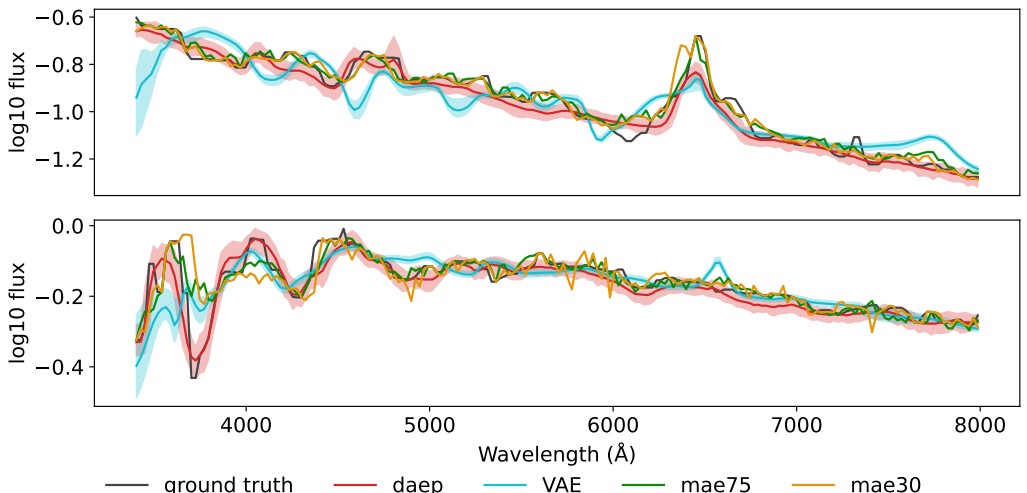

Figure 5: Reconstruction of SEDM spectra for ZTF supernovae from four latent tokens of dimension four using daep and baseline models.

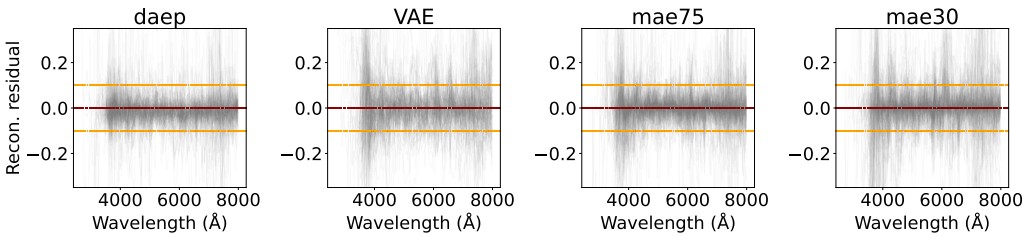

Figure 6: Residuals for test spectra from the ZTF BTS sample. daep achieves consistently lower reconstruction residuals across the considered wavelength range compared to the VAE and MAE baselines. Red line marked 0 residual while orange line marked at $\pm 0.1$.

**Downstream classification task.** Next, we perform linear probing on a three-way classification task to distinguish type Ia, type Ib/c, and other supernovae. For MAE, we unmask all measurements during probing. Results are reported in table 2. daep achieves both the highest classification accuracy and the highest $F_1$ score.

## 4.3 PHOTOMETRY OF SUPERNOVAE.

**Data source.** We used photometry from ZTF BTS (Bellm et al., 2018). Supernova flux was measured in two photometric filters or "bands" — green (g) and red (r) — along with spectra, all sampled

| Method | Abs. reconstruction error ↓ | Linear probing Accu. ↑ | Linear probing F$_1$ ↑ |
|---|---|---|---|
| daep | **0.040 (0.028)** | **0.82 (0.009)** | **0.45 (0.020)** |
| VAE | 0.070 (0.047) | 0.81 (0.002) | 0.40 (0.009) |
| mae-75% | 0.047 (0.026) | 0.79 (0.002) | 0.34 (0.010) |
| mae-30% | 0.066 (0.032) | 0.79 (0.001) | 0.30 (0.003) |

Table 2: Reconstruction and downstream classification metrics for ZTF BTS spectra. Reconstruction metrics are averaged over events; classification metrics are averaged over 10 probe/evaluation splits. Best models are boldfaced; underlined results overlap with the best mean within 1 standard deviation.

irregularly in time. We encoded this photometry (collectively denoted the supernova "light curve") into a two-token sequence of dimension two.

**Reconstruction.** In this task, we let the model reconstruct the observed data. For MAE-based models, we again provide 10% unmasked tokens during decoding. Example reconstructions are shown in fig. 7, with more in fig. 16. Residuals are plotted in fig. 8. daep achieves more accurate reconstructions than VAE, as also reflected in table 3. Interestingly, daep tends to overestimate brightness before peak ($< 0$ days), an effect not observed in other models, but still achieves the highest-fidelity reconstruction.

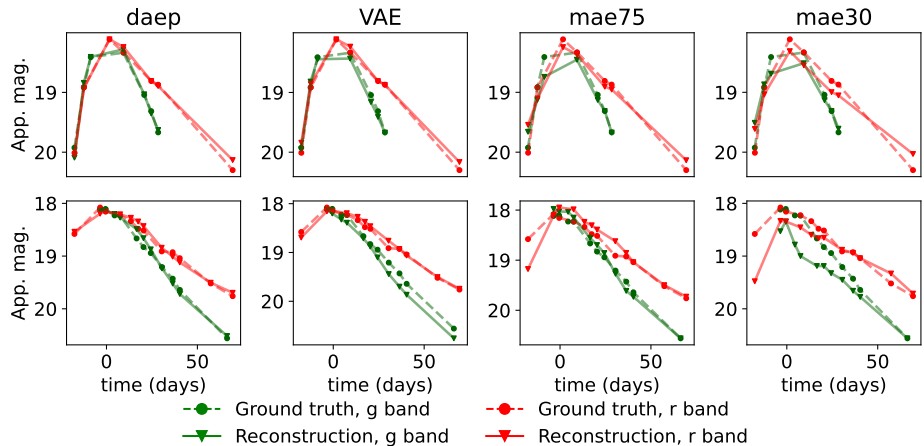

Figure 7: Light curve reconstruction from two latent tokens of dimension two using daep and baselines. (App. mag. stand for apparent magnitude).

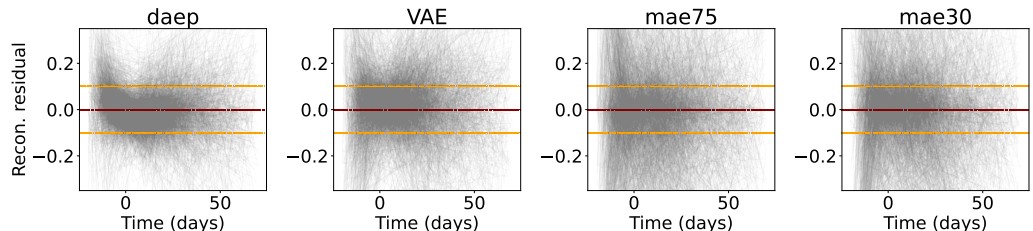

Figure 8: Residuals for test light curves from ZTF photometry. daep achieves smaller residuals overall than both VAE and MAE, but systematically overestimates the brightness before peak light. Red line marked 0 residual while orange line marked at $\pm 0.1$.

**Downstream classification.** We perform linear probing on a three-class task to classify supernovae as type Ia, type Ib/c, or other. For MAE, we use full measurements during encoding; other models use the latent. Results are shown in table 3. In this task, MAE achieves the best performance, likely because it better encodes general trends, while photometry contains fewer high-frequency features for daep to capture.

| Method | Abs. reconstruction error ↓ | Linear probing Accu. ↑ | Linear probing $F_1$ ↑ |
|---|---|---|---|
| daep | **0.078 (0.038)** | 0.82 (0.023) | 0.41 (0.061) |
| VAE | 0.11 (0.064) | 0.77 (0.040) | 0.41 (0.052) |
| mae-75% | 0.10 (0.083) | 0.88 (0.014) | 0.55 (0.024) |
| mae-30% | 0.091 (0.066) | **0.89 (0.003)** | **0.63 (0.044)** |

Table 3: Reconstruction and downstream classification metrics for ZTFBTS photometry. Reconstruction metrics are averaged over events; classification metrics are averaged over 10 probe/evaluation splits. Best models are boldfaced; underlined results overlap with the best mean within 1 std.

## 5 MULTIMODAL WITH MODALITY DROPPING

**Modality mixing and training for multimodal data.** To learn joint representations from multiple modalities, we use a late-mixing strategy. The goal is to have a latent representation that summarize all modalities in hand. This cannot directly be done with e.g., mixture of expert VAE or contrastive learning where each modality has own encoding. Each modality is first encoded with a perceiver encoder, a learnable modality embedding is then added to all tokens from that modality, and concatenated along the sequence dimension. A second perceiver encoder then acts as a "mixer" to produce a single compact bottleneck sequence. Because the perceiver encoder does not require fixed-length input, we employ modality dropping (Neverova et al., 2015; Liu et al., 2022) during training so the multimodal model can accommodate missing modalities. All modalities are decoded using modality-specific diffusion decoders. A schematic of this architecture with two modalities is shown in fig. 9.

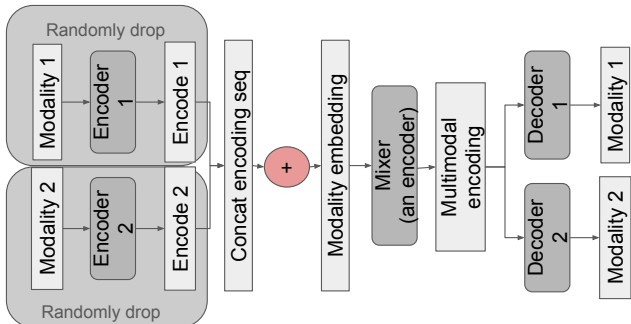

Figure 9: Late mixing and modality dropping for the multimodal daep model.

**Simulated supernova spectra and photometry.**

| Method | MSE -10 days | MSE 0 days | MSE 10 days | MSE 20 days | MSE 30 days |
|---|---|---|---|---|---|
| mmVAE | **0.080 (0.05)** | **0.0057 (0.005)** | **0.0053 (0.0051)** | **0.0064 (0.009)** | 0.1108 (0.021) |
| daep | 0.16 (0.10) | 0.017 (0.015) | 0.012 (0.011) | 0.011 (0.013) | **0.019 (0.027)** |
| contrastive | 0.55 (0.43) | 0.073 (0.032) | 0.10 (0.050) | 0.090 (0.048) | 0.11 (0.057) |

Table 4: Performance of the cross-modality inference task from photometry to spectra on the simulated dataset. We boldface the best-performing model and underline those whose 1 std interval contains the best mean. Our method performs similarly to mmVAE and outperforms contrastive search.

We use simulated type Ia supernova data from the radiative transfer models in Goldstein & Kasen (2018) and Shen & Gagliano (2025a). The dataset contains full spectral energy distributions for 5,000 simulated events. Shen & Gagliano (2025a) simulated idealized photometry with six filters from the Vera C. Rubin Observatory LSST (Ivezić et al., 2019). Each light curve is paired with five spectra taken at $-10, 0, 5, 10, 20$, and 30 days after peak brightness. We focus on the cross-modality inference task of reconstructing spectra from photometry. This task is a common one

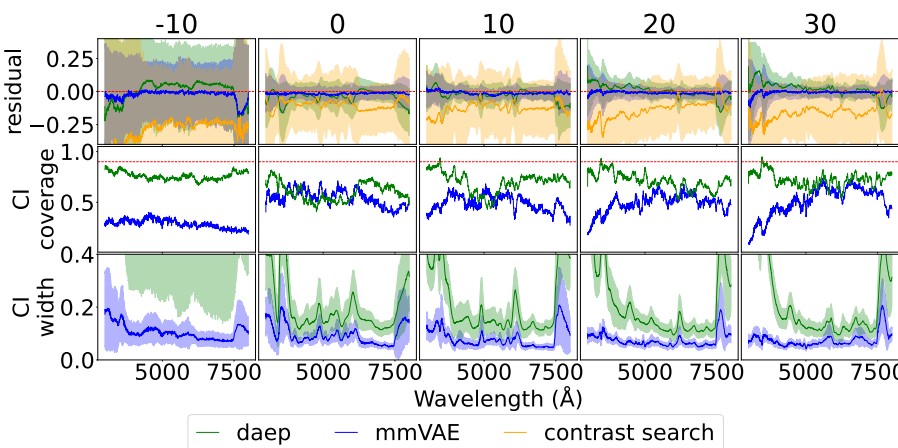

Figure 10: Performance of cross-modality inference with daep, mmVAE, and contrastive nearest-neighbor search. Our method achieves similar reconstruction to mmVAE, and better than contrastive search. Our method's CI widths exhibit better coverage than mmVAE at all phases relative to explosion.

in astrophysics: photometry is easy to obtain but spectra require significantly higher integration times, and spectroscopic datasets are therefore significantly more sparse. Since MAEs cannot handle completely missing modalities, we benchmark against (1) a mixture-of-experts VAE and (2) the contrastive-learning-based nearest-neighbor search from Shen & Gagliano (2025a).

We evaluate model performance using residuals, confidence interval (CI) coverage, and CI width as a function of wavelength and observation time, as shown in fig. 10 and table 4. Our method performs similarly to mmVAE, with slightly better coverage. Both substantially outperform contrastive search. This is likely because photometry contains less information than spectra, leading to modality collapse in weaker baselines (a similar conclusion is made by the authors of Zhang et al. 2024 using a comparable dataset).

## 6    DISCUSSION

In this work, we have presented an architecture for reconstruction-based SSL on multivariate sequential datasets. Although we validate daep primarily on astrophysical datasets, the architecture is domain-agnostic and directly applicable to other irregular, multimodal domains such as healthcare, finance, and sensor networks. The combination of perceiver tokenization and diffusion decoding enables scalable self-supervised learning on data that existing SSL methods cannot readily accommodate.

Our comparisons with perceiver-based MAEs reveal that decoder context plays a critical role: MAEs benefit from access to unmasked tokens, yet daep performs comparably or better without such context, and outperforms MAEs without decoder access. This highlights daep's ability to compress high-frequency information into a latent bottleneck. We also find that daep reconstructions more faithfully capture high-frequency spectral features, essential for enabling downstream tasks reliant on fine detail (e.g., the identification of short-duration signal anomalies).

While daep is more flexible than existing approaches, diffusion decoding is computationally heavier than masking, and our experiments focus on astronomy. Future work will broaden the evaluation to clinical, financial, and multimodal sensor datasets, and explore hybrid objectives that combine diffusion reconstruction with predictive or contrastive tasks (Huang et al., 2023) for better cross modality alignment, as well as using measurement noise aware losses. Beyond representation learning, daep may also serve as a generative model for simulating complex irregular multimodal phenomena and augmenting existing datasets. We plan to explore these extensions in future work.

## REPRODUCIBILITY STATEMENT

We detailed models and training in appendix A to reproduce our results. We further open our code in `https://anonymous.4open.science/r/Perceiver-diffusion-autoencoder-45B0`.

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

APPENDIX

# A IMPLEMENTATION DETAILS

## A.1 TOKENIZER DETAILS

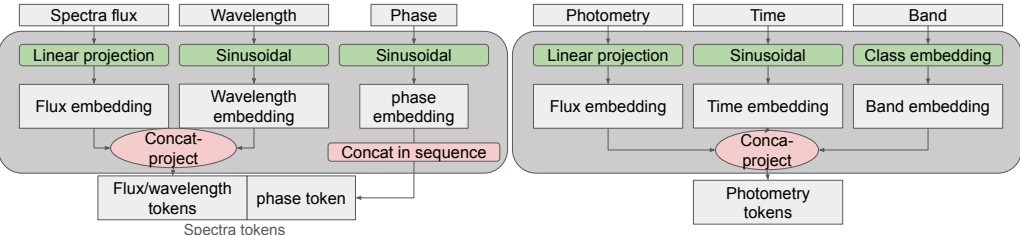

Figure 11: Tokenizers used in our empirical studies, from left to right: spectra (flux across wavelengths) and light curves (brightness in different colors over time).

## A.2 LAMOST MODEL DETAILS

**Data preprocessing** We enforced a 3-$\sigma$ quality cut, i.e., only measurements exceeding 3 times the measurment error are kept for modeling. After all quality cuts, we have 17,063 training stars. We took arcsinh of flux before modeling and after generation we calculate the original flux. Flux and wavelength are standardized to a z-score using the mean and standard deviation of the full training set after calculating arcsinh of flux.

**Architectural details Training details** We used a learning rate of $2.5 \times 10^{-4}$ for both models and

| model | bottleneck len | bottleneck dim | enc. layers | dec. layers | model dim | # heads | hidden seq len |
|-------|---------------|----------------|-------------|-------------|-----------|---------|----------------|
| daep | 4 | 8 | 4 | 4 | 128 | 8 | 256 |
| mae | 4 | 8 | 4 | 2 | 128 | 8 | 256 |
| VAE | 4 | 8 | 4 | 4 | 128 | 8 | 256 |

Table 5: Architectural choices used in LAMOST experiments.

trained for 2000 epochs and 200 epochs respectively for daep and VAE, confirming that the training loss converged for both models. We set $\beta = 0.1$ for the VAE.

## A.3 ZTF MODEL DETAILS

**Light curve preprocessing** We first enforced a 3-$\sigma$ cut on measurements, then used a Gaussian process to find the peak time of red band as the 0 phase We align time to be relative to the peak time. We only kept events whose light curves have measurement before and after the peak.

**Spectra preprocessing** We enforce a 3-$\sigma$ quality cut for both spectra and light curve, i.e., only measurements exceeding 3 times the measurement error are kept for modeling. After all cuts, we have 2,934 events left in the training set. We take the base-10 logarithm of the flux before modeling, and after generation we calculate the original flux. We also apply a median filter to filter out noise. Flux and wavelength values are then standardized to a z-score using the mean and standard deviation of the full training set.

**Architectural details**

**Training details** In contrast from our LAMOST experiment, we augment our data by 5 folds, adding noise to flux measurement and randomly masking measurements due to the small dataset size. We

| model | bottleneck len | bottleneck dim | enc. layers | dec. layers | model dim | # heads |
|-------|----------------|----------------|-------------|-------------|-----------|---------|
| daep  | 4              | 4              | 4           | 4           | 128       | 8       |
| mae   | 4              | 4              | 4           | 2           | 128       | 8       |
| VAE   | 4              | 4              | 4           | 4           | 128       | 8       |

Table 6: Architectural choices used in our ZTF supernova spectra experiments. We used a single-stage decoder (skipping the latent sequence) since the sequence is short.

| model | bottleneck len | bottleneck dim | enc. layers | dec. layers | model dim | # heads |
|-------|----------------|----------------|-------------|-------------|-----------|---------|
| daep  | 2              | 2              | 4           | 4           | 128       | 4       |
| mae   | 2              | 2              | 4           | 2           | 128       | 4       |
| VAE   | 2              | 2              | 4           | 4           | 128       | 4       |

Table 7: Architectural choices used in ZTF light curve experiments. We used a single-stage decoder (skipping the latent sequence) since the sequence is short.

used same learning rate of $2.5 \times 10^{-4}$ for both models and trained for 2000 epochs and 200 epochs respectively for daep and VAE, confirming that the training loss converged for both models. We set $\beta = 0.1$ for the VAE.

## A.4 MULTIMODAL SPECTRA AND PHOTOMETRY

**Data preprocessing.** We did not perform further processing beyond those described in Shen & Gagliano (2025b).

**Architectural details** We have the first stage encoder for both light curves and spectra to have a model dimension of 256, 4 layers, 4 heads, and 64 tokens after encoding. The modality mixer has 4 layers and 4 heads and model dimension 256, and during encoding we allow the concatenated sequence to attend to itself. We encode to a bottleneck sequence of 4 tokens of dimension 4 each.

**Training details.** In each batch, we randomly dropped each data modality with probability 0.2, making sure that at least one modality is retained. We trained with a learning rate $2.5 \times 10^{-4}$ and for 2000 epochs, confirming convergence of the loss.

# B  FURTHER EXPERIMENTAL RESULTS

## B.1  LAMOST SPECTRA

In fig. 12, we show additional spectra reconstructions using daep and VAE baselines. Our method consistently captures higher-frequency information details compared to the VAE baseline.

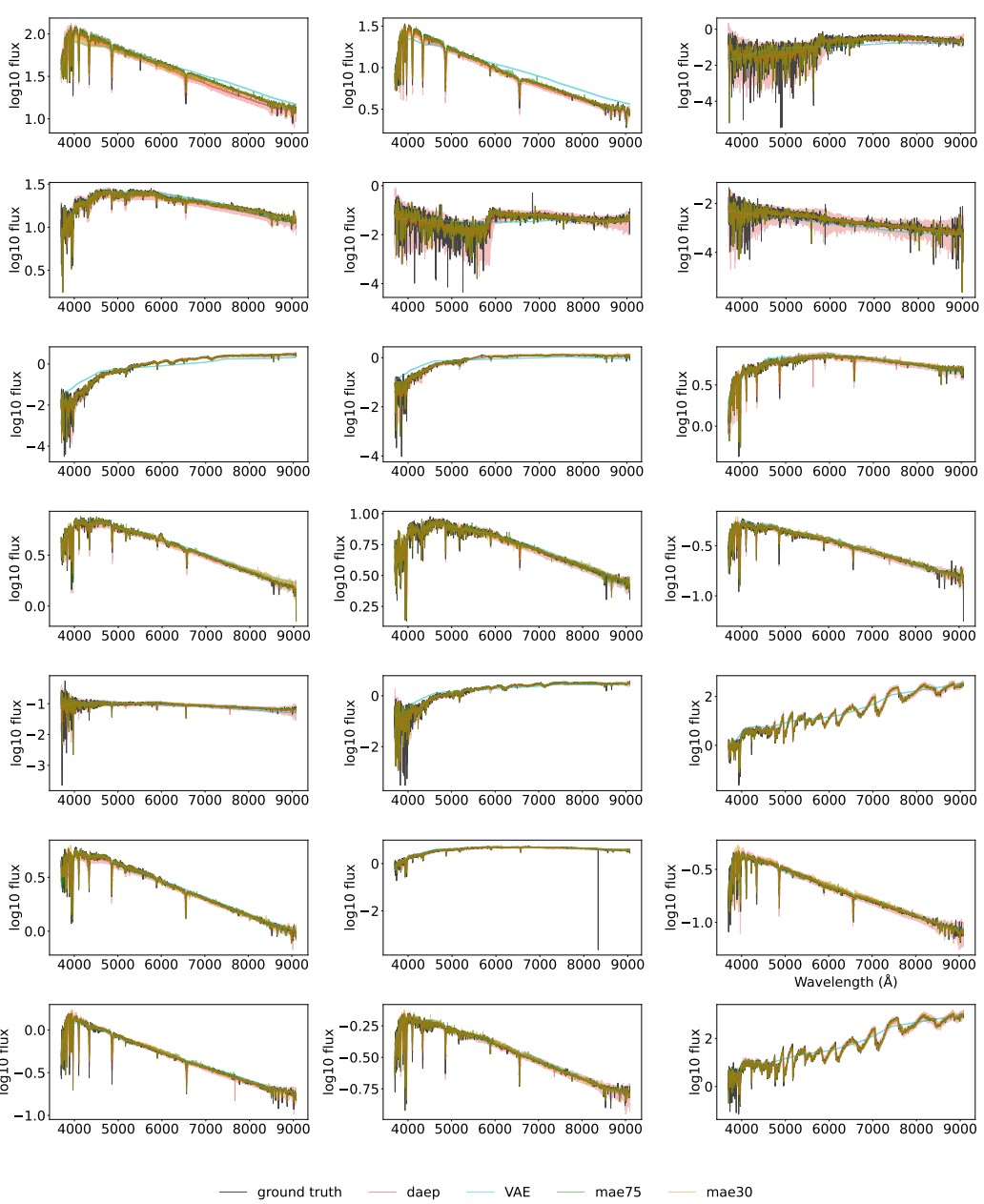

Figure 12: Reconstructions of additional LAMOST variable star spectra. Our method (red) captures more high-frequency absorption features than the VAE baseline with the same-sized bottleneck representation (blue).

In fig. 13, we compare the latent representations of LAMOST spectra (after t-SNE) from daep, mae and VAE, colored by variable star classification.

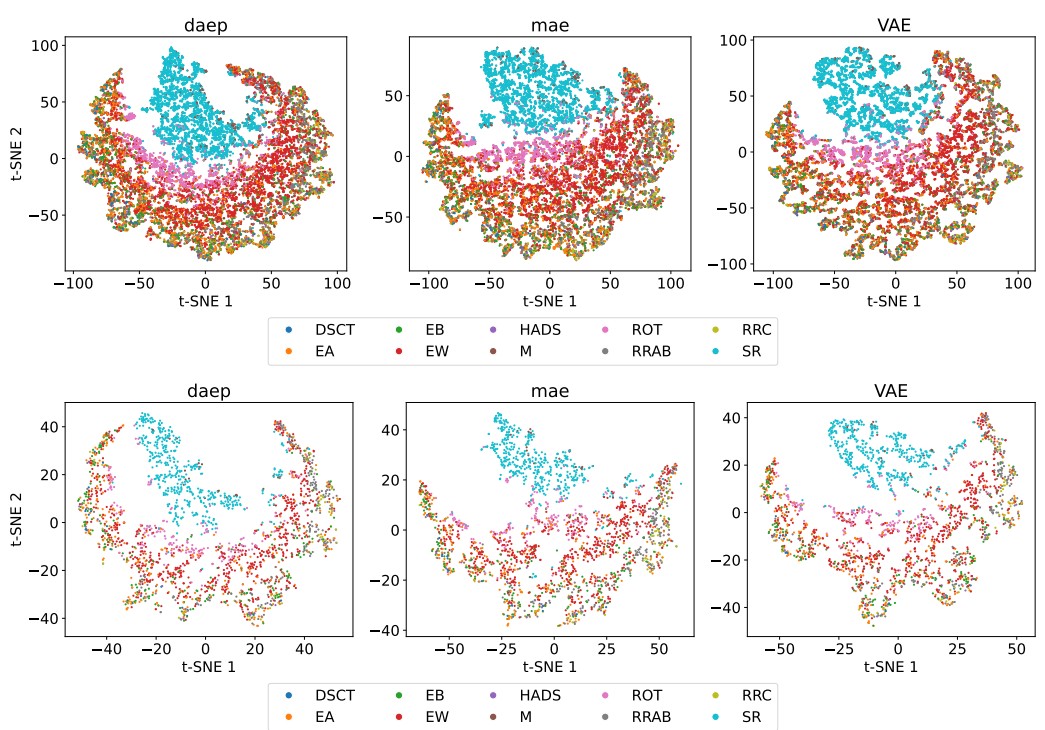

Figure 13: Latent representations of LAMOST spectra from training (upper) and test (lower) sets. The latent space for daep appears more well-regularized compared to a VAE of comparable dimensionality.

## B.2 ZTF SPECTRA

In fig. 14, we show additional spectroscopic reconstructions for ZTF BTS supernovae. Our method captures finer details and produces better-covered posteriors than the VAE baseline.

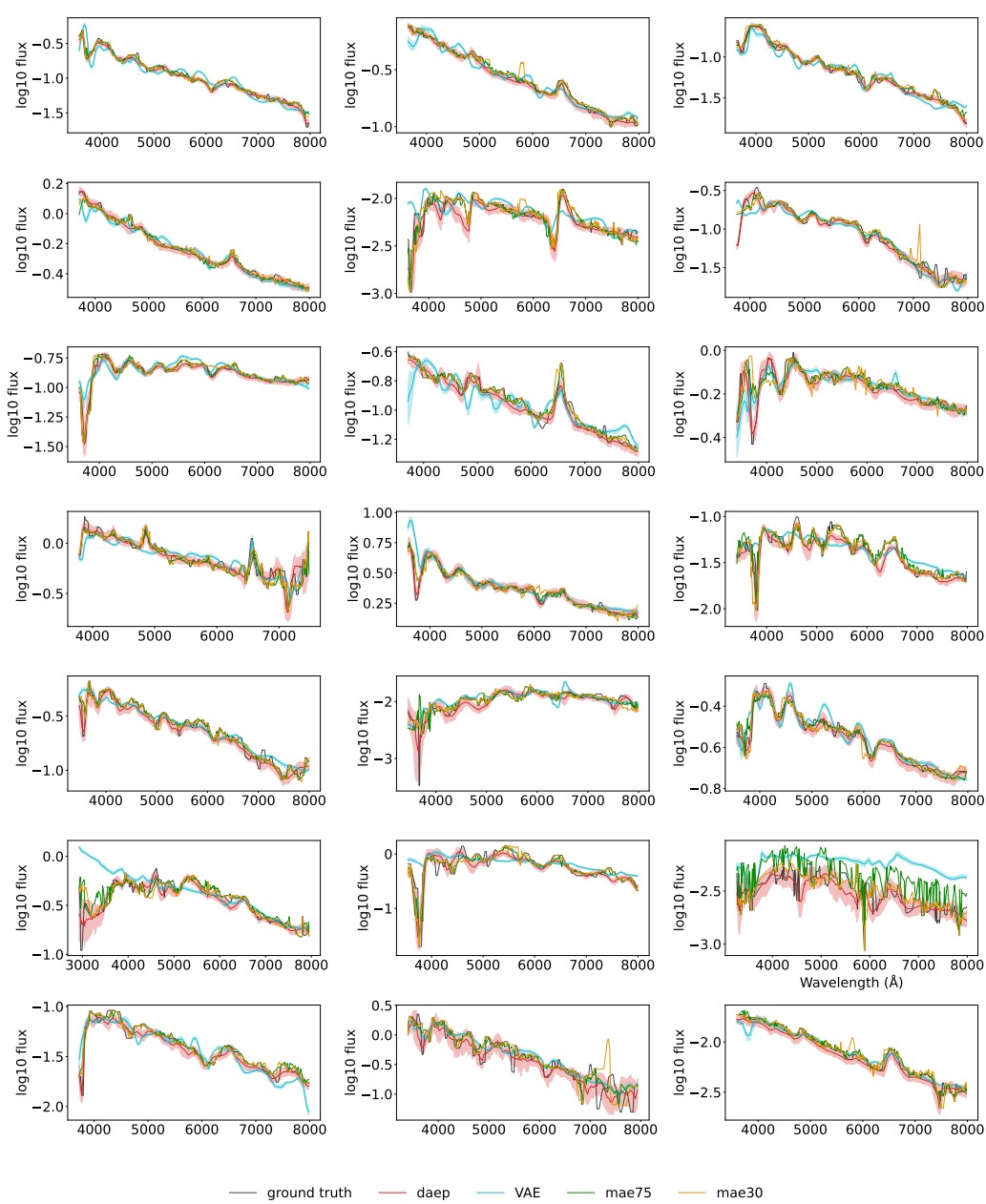

Figure 14: Additional ZTF spectra reconstructions with daep and VAE. Our method captures finer details and maintains better posterior coverage.

In fig. 15, we compare latent representations of the ZTF spectra (after t-SNE) from daep, mae and VAE, colored by event type. Interestingly, the daep latent space appears more continuous than either the MAE or the VAE with $\beta = 0.1$.

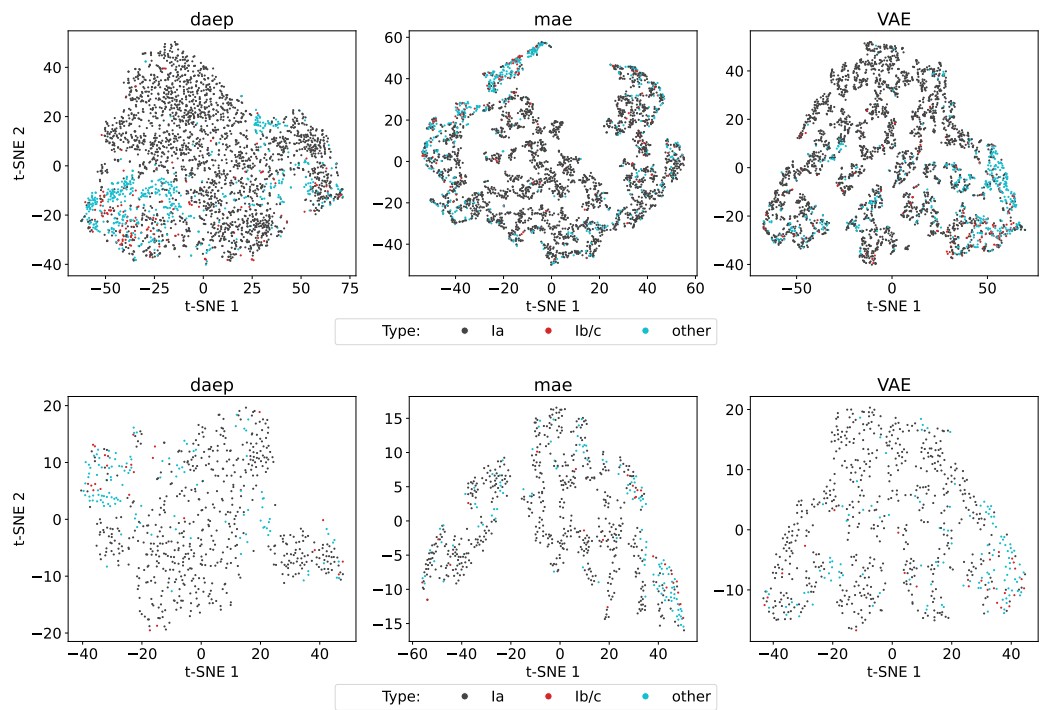

Figure 15: Latent representations of ZTF spectra from training (upper) and test (lower). The latent space for daep appears more well-regularized compared to a VAE of comparable dimensionality.

## B.3 ZTF LIGHT CURVES

In fig. 16, we show a series of ZTF light curve reconstructions for daep alongside the VAE and MAE baselines. Our method performs superior reconstructions compared to the baseline models, particularly the MAE with 75% of the input data masked.

In fig. 17, we show the light curve latent representations after t-SNE for all three models considered in this work. As with the ZTF spectra, daep's latent space appears more continuous than either MAE/VAE baselines (though type Ib/c supernovae do not appear as well-separated as in the MAE space, as indicated by the higher mean $F_1$ score from mae30 listed in table 3.).

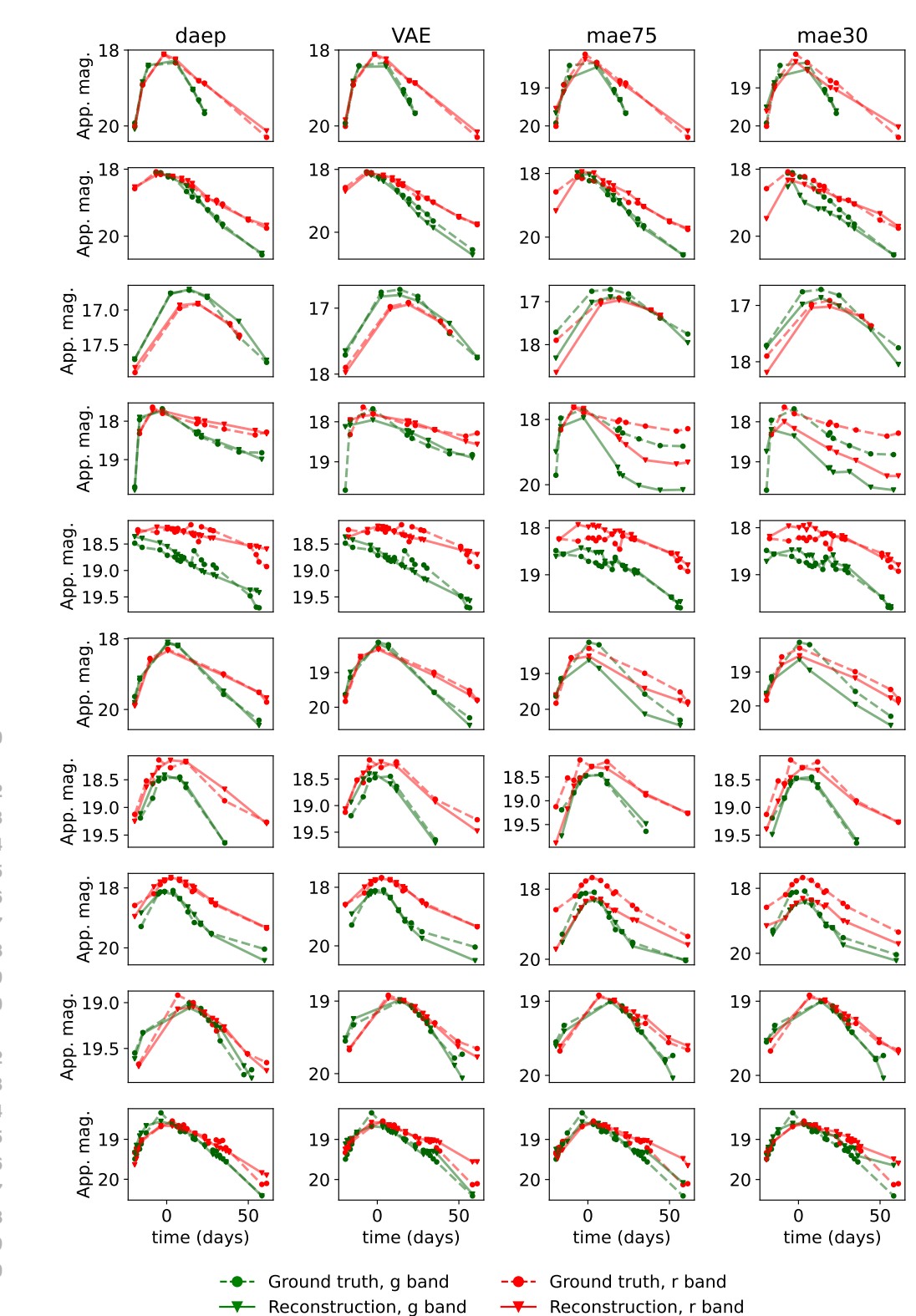

Figure 16: Additional examples of ZTF light curve reconstructions.

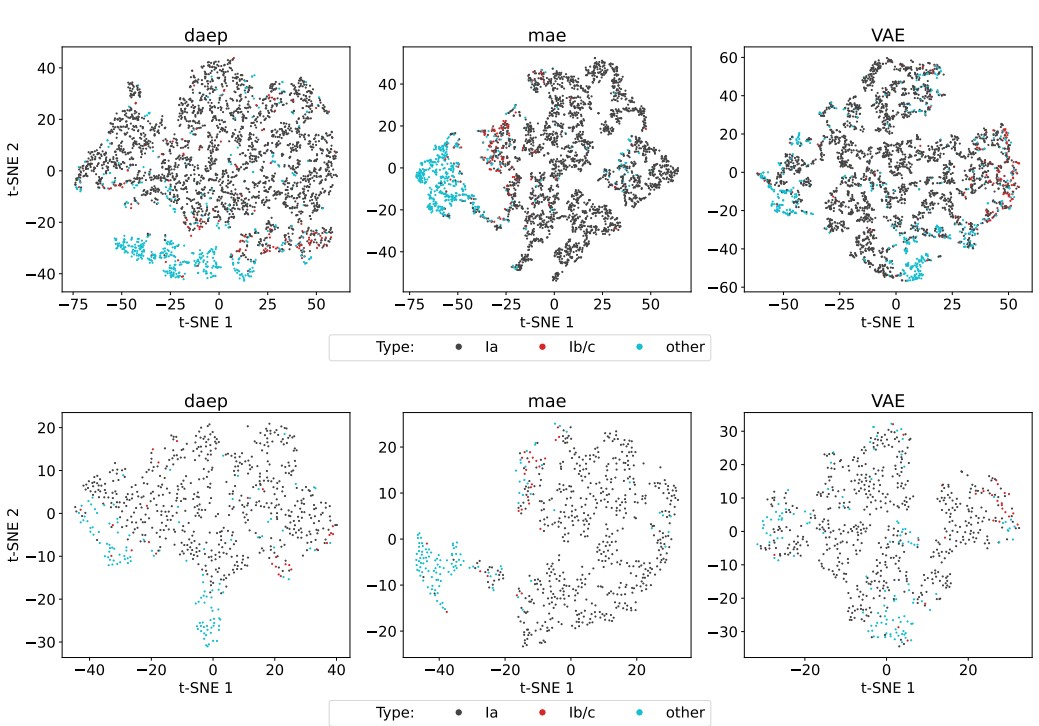

Figure 17: Latent representations of ZTF photometry from training (upper) and test (lower). The latent space for daep appears more well-regularized compared to a VAE of comparable dimensionality.

# C    USE OF LARGE LANGUAGE MODELS (LLM)

We used LLM (ChatGPT) for grammar checks and polishing only.

