# OpenReview forum: "Diffusion Autoencoders with Perceivers for Long, Irregular and Multimodal Sequences"
_ICLR.cc/2026/Conference — Submitted to ICLR 2026_

### Official Review · Reviewer_xLSY · 2025-10-28

**Soundness:** 3
**Presentation:** 3
**Contribution:** 2
**Rating:** 4
**Confidence:** 3

**Summary:**

The paper proposes daep, a diffusion-autoencoder built incorperating perceiver layers to handle multimodal, irregularly sampled astronomy data.

**Strengths:**

The (value, position, metadata) tokenization is clean and extensible, supporting both continuous positions (sinusoidal embeddings + MLP) and discrete positions (learned class embeddings), while metadata are naturally incorporated as additional tokens. Aggregating variable-length sequences into a fixed set of latent vectors via cross-attention in Perceiver layers is an elegant solution for the irregular astronomical data considered here. The decoder mechanism for generating variable-length predictions could be clarified in the presentation, though it appears to work by querying at the specific positions to be evaluated.

The introduction effectively motivates the work by highlighting irregular, long, and multimodal sequences as common challenges across biomedicine, finance, and astronomy. While current experiments focus on astronomical data, the architectural contributions are relevant to a wider range of domains.

The authors provide public code, detailed training specifications, and complete architectural configurations for each dataset. These details significantly lower the barrier for future work building on this method.

**Weaknesses:**

My main concern with this work is that it does not compare against the most natural alternative architectures for handling long sequences. In particular, substituting the Perceiver layers with modern long-context sequence models such as Mamba, xLSTM, DeltaNet, or RWKV seems like a minimal and necessary baseline to contextualize performance. Given that the paper’s central claim is to provide a general architecture for long, irregular, multimodal sequences, limiting baselines to Perceiver-family MAE and (β-)VAE makes it difficult to assess the relevance and strength of the proposed approach.

The idea of applying attention-based architectures to irregular data is not new (e.g., Horn et al. 2020, https://proceedings.mlr.press/v119/horn20a.html). Combined with the astronomy-specific evaluation, the framing of the paper as a general method feels overstated. A broader or more diverse set of benchmarks would be required to support the current framing convincingly. Key dataset sizes are modest (e.g., LAMOST: 17,063 train / 2,225 test) and some ZTF settings require augmentation due to small size. This raises questions about scalability and performance on larger, more complex datasets. Expanding the empirical evaluation or clearly reframing the paper as focused on astronomy applications would strengthen the contribution.
Underwhelming multimodal performance.

Minor concerns:

In the multimodality section, the proposed method does not clearly outperform simple alternatives. On the cross-modality task (photometry → spectra), mmVAE performs best at all but one phase (even if confidence intervals overlap). It is unclear whether daep would hold up against stronger baselines specifically designed for this setting.
Presentation clarity could be improved.

The architectural innovations and the interactions between components are currently described mostly in free text, which makes it harder to follow. Figure 2 in particular could be more clearly labeled and captioned to make the data flow, input/output interfaces, and conditioning points explicit. A more structured and annotated diagram would substantially improve readability and reproducibility. It would be helpful to provide, e.g., tensor shapes and other information to help disambiguating the different inputs/outputs. More details on how to perform inference with the proposed model would be helpful as well.

**Questions:**

Please consider adding comparisons to some of Mamba/state-space models, xLSTM, DeltaNet/RetNet-style kernels, and RWKV on the same tasks, ideally with comparable parameter counts and context budgets.

Why do you think does mmVAE seemingly outperform daep at early phases?

It would be informative if you reported additional training details such as  training/inference wall-clock, number of GPUs, GPU type and GPU hours used to train. Also how does the proposed architecture scale to bigger datasets and bigger model parameter counts?

How to use the provided repository? There is no readme.md

---

### Official Review · Reviewer_P1L6 · 2025-10-30

**Soundness:** 3
**Presentation:** 3
**Contribution:** 1
**Rating:** 2
**Confidence:** 4

**Summary:**

This work introduces daep, an application of the Perceiver architecture to diffusion autoencoders. Perceivers naturally handle irregularly sampled sequences, making daep a well-motivated approach to SSL for irregularly sampled inputs. Evals are performed on 3 different unimodal astronomical datasets/tasks, as well as a multimodal use case.

**Strengths:**

This paper pinpoints and attempts to address an important topic in real-world and scientific data. Evaluations are performed on a variety of sensible baselines and datasets on both pretraining and downstream task performance.

**Weaknesses:**

- Main issue: While daep reconstruction certainly appears more faithful than the baseline methods, improvement in downstream task accuracy overall does not seem as significant (generally <=1% absolute or worse than baselines). I would recommend the authors find downstream tasks that better demonstrate daep's unique advantages over other methods: since this is a representation learning method, the main payoff is on downstream tasks of interest and the minimal improvements demonstrated so far are not so convincing.
- Relatedly, a discussion on the computational requirements of daep compared to simpler methods (e.g., MAE) should be added, especially since the improvements in downstream task performance are mostly incremental.
- How is the std calculated? How many seeds? For the linear probing std results, is it n linear probing runs with different seeds on top of the same pretrained model or n separate pretraining runs?
- Line 219: what does "linear probing on 30% of the test set" mean? What is used to train the linear probe? What is used as a test/validation set?
- It would be interesting to see a comparison with vanilla diffusion AE, taking into account the irregular sampling through positional encoding. However, I would recommend addressing the "main issue" above first before spending a lot of time on this during the rebuttal period.
- It would also help to do an ablation study on the size of the latents used. e.g., maybe the VAE performance would improve a lot with slightly more capacity in the latent space.
- The related works section should include a discussion on techniques to encode irregularly sampled sequences (perhaps including applications of these to astronomical/other scientific data).
- There should be an appendix section with details about the datasets used. e.g., daep's improvement in F1 score in section 4.2 is much more prominent than in accuracy. is this because the dataset is very imbalanced?

**Questions:**

see 'weaknesses'.

---

### Official Review · Reviewer_5UWJ · 2025-10-31

**Soundness:** 2
**Presentation:** 2
**Contribution:** 2
**Rating:** 4
**Confidence:** 1

**Summary:**

In this paper, the authors propose a diffusion-based representation learning method for long, irregular sequences, namely **daep**. Specifically, they encode the input into a conditioning vector and train a conditional diffusion model with these encoded conditionings as a decoder, mimicking the setup of diff-AE [1]. To be compatible with sequences, they also propose a tokenizer that can handle generally irregularly measured sequences. In the decoder, the authors use *Perceiver* modules by adding a second cross-attention whose queries carry positional information.

To validate the effectiveness of their method, the authors compare daep to VAEs and MAEs, showing that daep achieves lower reconstruction error and more discriminative latent spaces. The experiments are mainly conducted on astronomy datasets, with reconstruction, classification, and cross-modality inference tasks.

**Strengths:**

1. Enables a Perceiver encoder that flexibly handles variable-length tokenized inputs across modalities, with technical improvements that adapt diff-AE to long-sequence processing.
2. daep’s performance appears more stable, while MAE is sensitive to the masking ratio; this is verified across a range of astronomy datasets.

**Weaknesses:**

1. Some claims may be overstated. For example, “diffusion models’ latent spaces can be elusive” (line 50) is not fully convincing, as there is a strong line of work showing diffusion representations are informative and discriminative [2,3]. For instance, linear probing with DiT intermediate representations yields near-SoTA accuracies [4].
2. The main novelty seems to be adapting the diff-AE framework to sequence processing. Perceiver blocks instantiate an extra cross-attention from a new query set (with positional information). This is not novel in diffusion models; prior work also utilizes additional cross-attention modules to introduce auxiliary information [5,6].
3. Experiments: the authors only evaluate and compare Perceiver-based models. Are there stronger baselines? I also suggest reporting training cost (time/FLOPs/parameters) for daep and other models, since daep is likely more computationally heavy than MAEs. Ablations on key components (e.g., the cross-attention/Perceiver choice) would better motivate the design.

**Questions:**

1. The paper aims to learn very low-dimensional representations (e.g., dim=4 in LAMOST dataset) on relatively small datasets. Given that DiT architectures are typically used at scale, what motivates a very small DiT here, and is this necessary/feasible?
2. The figures (e.g., Fig. 1 and 2) are not very illustrative. Please consider polishing them for clarity (module boundaries, token shapes, and data flow).

*My current rating is not a final assessment, as I am not very familiar with astronomy. I may revise my rating after discussion with the authors and other reviewers.*

**References**

 [1] Preechakul, Konpat, et al. *Diffusion autoencoders: Toward a meaningful and decodable representation.* CVPR, 2022.
 [2] Xiang, Weilai, et al. *Denoising diffusion autoencoders are unified self-supervised learners.* ICCV, 2023.
 [3] Li, Xiao, et al. *Understanding representation dynamics of diffusion models via low-dimensional modeling.* NeurIPS, 2025.
 [4] Chen, Xinlei, et al. *Deconstructing Denoising Diffusion Models for Self-Supervised Learning.* ICLR, 2025.
 [5] Nichol, Alex, et al. *GLIDE: Towards photorealistic image generation and editing with text-guided diffusion models.* ICML, 2022.
 [6] Hertz, Amir, et al. *Prompt-to-Prompt Image Editing with Cross-Attention Control.* ICLR, 2023.

---

### Official Review · Reviewer_q6xq · 2025-10-31

**Soundness:** 3
**Presentation:** 2
**Contribution:** 3
**Rating:** 4
**Confidence:** 4

**Summary:**

The authors develop daep, a model for autoencoding irregular, multimodal sequential data. It is in essence a Perceiver IO model, but with a diffusion decoding head. They show that this model outperforms MAE and VAE models on reconstruction and classification tasks in astronomy. They also show that a multimodal variant of daep performs similarly to baselines.

**Strengths:**

This is a nice work presenting an interesting approach to tackling irregular sequential data. The topic is a very pressing one, and this is a nice contribution to the area. This approach of combining the Perceiver model with a diffusion decoder is novel, and both methods have separately been shown to work well in many other settings. The evaluations are reasonable and performance is shown to be strong in a variety of settings.

**Weaknesses:**

MAE: I am slightly confused about how this works. As I understand it (let me know if I am misunderstanding something), the encoder takes in the full input into the encoder to produce a latent representation, and then the decoder takes in that latent representation, the full input which has now been masked at some masking fraction, and a query with is the positions of the masked positions. Why do the authors provide 10% of the inputs for free during reconstruction? I would appreciate more clarity in the explanation.

Section 4.1: Absolute reconstruction error is fine when you don’t have uncertainties on your measurements, but when you do (and you should have them for LAMOST spectra) it is much more typical, and meaningful, to report errors in standard deviations/sigmas. As in, instead of |measurement - reconstruction|, use |measurement - reconstruction| / measurement_error. Right now it appears that the all models perform reasonably well, but it is hard to compare, especially visually. Maybe the measurement uncertainty is extremely small on the spectra, and the reconstructions are actually not doing so well. It is unclear whether this is the case at the moment.

Figure 3: What is the dark red line (clearly visible in the inset)? This is not in the legend. daep appears to be pink in the legend? The thick lines also make it extremely difficult to compare the performance of the different models as they overlap heavily. I would like to see this with much thinner lines.

Figure 4: What are the units on the y axis here? Again if this is in absolute flux, I would like to see this in standard deviations.

Sections 4.2, 4.3: All the same questions/concerns/requests as in Section 4.1.

Section 5: Could the authors provide an explanation of, or a citation for mmVAE? I am not familiar with this model and it does not appear to be described anywhere in the text.

**Questions:**

L85-86: “Because the encoding guides every denoising step, diffusion autoencoders capture fine-grained detail more effectively than, for example, variational autoencoders”. Do you have a source for this? I don’t think this is the reason why diffusion autoencoders capture fine grained detail more effectively than VAE. It is that VAEs effectively learn the mean of the distribution because you are training with a pixel-space L2 loss. If you drew samples from a diffusion model and took the mean they would look blurry like VAE samples (no fine-grained detail). It doesn’t have to do with the conditioning.

L93-94: Is the citation to Dhariwal & Nichol, 2021 supposed to be a different citation? That is not the diffusion transformers paper.

Section 4.1: To clarify, do the authors undo the z scaling and arcsinh transform for the final output?

Figure 3: “the VAE mainly reproduces the continuum, likely due to posterior collapse.” Again I believe that it is just doing that because you have trained it with an L2 loss.

Table 1: Just out of curiosity, what accuracy would a naive baseline that always predicts the most common class achieve for the accuracy here? (or: what percentage of the test set is made up of the largest class?)

Table 4: Do the authors know why mmVAE outperforms daep for the cross-modality inference task for all cases except for the 30 day case where it is significantly worse than daep?

---

### Meta-Review · Area_Chair_MqFe · 2026-01-02

**Summary:**

All 4 reviewers recommend rejection. There is no rebuttal. The AC sees no basis to overturn the reviews.

**Reviewer Concerns:**

Not applicable. There is no rebuttal.

**Reviewer Scores:**

Not applicable. There is no rebuttal.

---

### Decision · Program_Chairs · 2026-01-26

Reject